# Soil Nutrient, Salinity, and Alkalinity Responses of *Dendrocalamopsis oldhami* in High-Latitude Greenhouses Depending on Planting Year and Nitrogen Application

**Zixu Yin** [1,2], **Xiao Zhou** [1], **Dawei Fu** [1], **Xuan Zhang** [1], **Liyang Liu** [1], **Zhen Li** [1] **and Fengying Guan** [1,2,*]

[1] International Center for Bamboo and Rattan, Key Laboratory of National Forestry and Grassland Administration, Beijing 100102, China; yinzx1994@126.com (Z.Y.); zhouxiao@icbr.ac.cn (X.Z.); zhangxuan@icbr.ac.cn (X.Z.); captaina1997@163.com (L.L.); captaina1998@163.com (Z.L.)

[2] National Location Observation and Research Station of the Bamboo Forest Ecosystem in Yixing, National Forestry and Grassland Administration, Yixing 214200, China

[*] Correspondence: guanfy@icbr.ac.cn

**Abstract:** This study explored the viability of greenhouse cultivation of *Dendrocalamopsis oldhami* under the "South Bamboo North Transplanting" initiative. In this study, the effects of planting year and nitrogen application on changes in soil nutrient levels, salinity, and alkalinity over the plant growth period were explored. After the introduction and planting of bamboo in 2017, a soil layer with a thickness of 0–40 cm was sampled at the end of the shooting stage in the greenhouse between 2017 and 2019 (late August), and the bamboo shoot yield and standing culm density were measured. Following the application of nitrogen to the bamboo groves in 2019, three nitrogen levels were established: no nitrogen ($N_1$:0 g grove$^{-1}$), medium nitrogen ($N_2$:540 g grove$^{-1}$), and high nitrogen ($N_3$:1080 g grove$^{-1}$). Soil layers at depths of 0–20 and 20–40 cm were sampled during the shoot elongation stage (late May) and at the end of the shooting stage (late August). The yield and nutrient content of bamboo shoots under different nitrogen treatments were also investigated. The results showed that $Ca^{2+}$ and $HCO_3^-$ were the main salt ions in greenhouse soil. With later planting years, the total number of cations ($Ca^{2+}$, $Na^+$, $Mg^{2+}$, and $K^+$) decreased, whereas the total number of anions ($HCO_3^-$, $SO_4^{2-}$, $NO_3^-$, and $Cl^-$) increased, resulting in a decrease in the percentage of exchangeable sodium (ESP), pH, and electrical conductivity (EC). The diameter at breast height, individual weight, and quantity of bamboo shoots increased annually, and the standing culm density increased by 1.4 times. Each year, the total nitrogen content decreased, whereas the alkali-hydrolyzed nitrogen, available phosphorus, and available potassium contents increased. Nitrogen application resulted in a significant decrease in ESP and pH and an increase in the total anion, cation, and EC values. It also reduced soil organic carbon, total nitrogen, total phosphorus, total potassium, available phosphorus, and available potassium. Nitrogen application increased the number of bamboo shoots, total yield, and accumulation of N and P; however, there was no significant difference between $N_2$ and $N_3$. In conclusion, the salinization of calcareous soil was alleviated, and the available nutrients were activated following the introduction of *D. oldhami* from south to north. The mineralization rates of organic matter and soil fertility increased. Soil acidification and EC decreased at the end of the shoot stage. Nitrogen application acidified the soil, and the yield and soil salt accumulation increased with increasing nitrogen levels. The nutrient uptake efficiencies of nutrients at high nitrogen levels were lower than those at medium nitrogen levels. Therefore, soil salt concentrations with values $0.26 < EC < 0.42$ hindered the nutrient uptake of *D. oldhami*.

**Keywords:** planting years; nitrogen; ESP; pH; salt ions; nutrient



## 1. Introduction

Global warming is expected to increase agricultural production at higher latitudes [1]. Higher temperatures at middle and low latitudes prolong the planting time of crops [2].

In the context of environmental changes and the possibility of artificial plant breeding, the range of crops traditionally grown at low latitudes can diversify to grow at high latitudes. Tropical resources from low latitudes contain abundant high-quality germplasm, demonstrating pest and drought resistance and infertility tolerance [3]. The use of tropical germplasm resources in temperate regions is important for coping with warming climates and addressing food security issues [4,5]. Bamboo is an important forestry resource. However, only a few high-latitude bamboo species exist. Since the 1980s, as a result of the "South Bamboo North Transplanting" project, the number of ornamental bamboos that exhibit strong cold resistance, such as *Phyllostachys aureosulcata*, *Phyllostachys vivax*, *Phyllostachys angusta*, *Phyllostachys bissetii*, *Phyllostachys propinqua*, *Indocalamus longiauritus*, and *Shibataea chinensis*, and which can be cultivated at high latitudes, has gradually increased in China [6,7]. The original sympodial bamboo from subtropical and tropical areas presents weak cold resistance, and the natural conditions in high-latitude areas make survival difficult. Generally, bamboo is planted in artificial greenhouses, where warm and humid conditions can be provided and is suitable for growth in weakly acidic soils. However, soil in the arid and semi-arid regions of northern China mostly consists of calcareous soil, which is characterized by a saturated base, neutral-to-alkaline acidity, and low availability of soil nutrients [8]. When conducting cultivation in artificial greenhouses, soil salinization is a serious issue because of the high input of water and fertilizer, the high-intensity utilization, the high temperature and humidity, the lack of natural rain, and other special conditions [9–11].

Saline–alkaline soil is a land type widely distributed in China and includes solonchak, solonetz, salty soils, and alkalized soil. These soils primarily contain potassium, calcium, sodium, magnesium chloride, sulfate, and bicarbonate. The type and quantity of soil salt ions in greenhouses lead to differences in the surface charge, pH, and exchangeable bases of the soil. Different studies have reported different results regarding the key salt ions that cause soil salinization [9,12–16]. Sun et al. [17] used electrical conductivity (EC) as an evaluation index to investigate the effects of soil salinization. With an increasing number of planting years in a greenhouse, there is a considerable change in soil salt ion composition, resulting in a serious loss of soil mineral nutrient elements, acidification of the soil, and decreased soil fertility [14,15,18–24]. Nitrogen is the most widely used fertilizer in greenhouse cultivation, and it is also the nutrient that bamboo requires the most. Urea, an amide nitrogen, cannot be directly absorbed by plants from the soil. However, soil microorganisms perform the vital task of converting it into nitrate or ammonium nitrogen, which can be absorbed by plants., As a neutral fertilizer, urea is versatile and suitable for various soils and plants. Moreover, it is easy to store and apply and has a minimal impact on soil quality. Urea is the most abundant nitrogen fertilizer and is also a widely used chemical nitrogen fertilizer. Some studies have suggested that appropriate nitrogen application could enhance the tolerance of plants to salt [25,26]. However, low nitrogen efficiency, serious nutrient loss, and poor crop quality are significant problems when cultivating from saline–alkaline soils [27–29]. Bamboo is regarded as an invasive species because of its strong reproductive ability, rapid growth rate, and intertwined underground rhizome roots, which intertwine with each other. Following the introduction of bamboo, the soil ecology of mixed forests can easily change, including soil pH, biodiversity, enzyme activity, and nutrient composition [30–35]. Bamboo is not a halophyte; however, there are reports of its planting on saline–alkali land, and it is believed that bamboo plants have some saline–alkali tolerance. Its introduction can also improve the ecological restoration, forest structure, and diversity of tree species for ecological restoration in coastal saline–alkali lands [36–38]. However, studies on bamboo in saline–alkali land have mainly focused on the effects of plant physiology and growth and standing forest, with few studies investigating changes in soil properties. *Dendrocalamopsis oldhami* is a sympodial bamboo, with shoots (called "fruit shoots") that are rich in nutrients, sweet to taste, and of considerable importance in resource development and utilization. A comprehensive test analysis of the salt tolerance, wind resistance, productivity, bamboo shoot quality, and economic value of *D. oldhami*

on a coastal mudflat revealed that *D. oldhami* is tolerant to salt and alkali conditions and can be planted on saline–alkali land with a salt content below 0.41% [39]. In this present study, *D. oldhami* was introduced to high latitudes (Beijing) for the first time and tested in a greenhouse. The shooting stage of *D. oldhami* in facilities has significantly extended. The shoot elongation stage occurs from April to mid-May. Many bamboo shoots were concentrated in summer (July–August), with a decrease in temperature during autumn and winter (September–December); however, the amount was very small. Problems such as plant height, cold tolerance, and light adaptability have been explored previously [40–42]. However, we found that the quality of *D. oldhami* shoots was poor after introduction, and the weight and total yield of the bamboo shoots were unsatisfactory, which were speculated to be caused by the nature of the calcareous soil in the greenhouse. From the perspective of soil adaptability, the soil acid–base properties, nutrients, and salt content with planting years and nitrogen application levels of *D. oldhami* in a high-latitude greenhouse were studied. The effects of planting years and nitrogen fertilizer levels on salinity and soil fertility were analyzed to provide a theoretical basis for scientific soil management in greenhouses and the sustainable management of *D. oldhami*.

## 2. Materials and Methods

### 2.1. Site Description

The experiment was conducted in a Xiaotangshan solar greenhouse in Beijing (116°23′ E, 40°22′ N). The average annual temperature outside the greenhouse is 11.8 °C, with the lowest temperature reaching as low as −16.8 °C, and the average annual precipitation is 550 mm. The greenhouse was able to maintain an average temperature of 17.4 °C, with the lowest temperature recorded at 0.7 °C. This temperature drop was observed during the early morning hours of 6:00–8:00 a.m. in December or January, which falls within the low-temperature tolerance range (−4 °C) of *D. oldhami*. The traditional arch-structured solar greenhouse covers an area of 1200 m$^2$ (3.5 m high back wall).

In April 2017, *D. oldhami* was introduced to the greenhouse; its provenance was in the Fujian Province. In August of the same year, newly grown bamboo shoots sprouted, and all of them were kept except for those that were diseased. In 2018, dead parent bamboo was removed, healthy new shoots from that year were used for replanting, and two to four shoots from the following year were maintained. The spacing between each clump was 1.5–2 m. The temperature, soil temperature, and soil humidity during the 2019 fertilization test are listed in Table 1. An automatic meteorological station (FRT-X06A, Fuotong Technology (Beijing) Co., Ltd., Beijing, China) was used for monitoring.

**Table 1.** Overview of air temperature, soil temperature, and humidity in the greenhouse from April to August 2019.

| Average Temperature (°C) | Maximum Temperature (°C) | Minimum Temperature (°C) | Soil Temperature (°C) | | | Soil Humidity (%) | | |
|---|---|---|---|---|---|---|---|---|
| | | | 0−15 cm | 15−30 cm | 30−45 cm | 0−15 cm | 15−30 cm | 30−45 cm |
| 23.86 | 41.70 | 1.20 | 22.79 | 22.14 | 21.26 | 60.48 | 63.64 | 69.35 |

### 2.2. Experimental Design

Before planting *D. oldhami* in April 2017, 1 kg of organic fertilizer (at least 40% organic matter; Beijing Yite Agricultural Technology Extension Service Co. Ltd., Beijing, China) was applied to each bamboo grove. Subsequently, no fertilizer was applied until March 2019. Three nitrogen fertilizer levels were established [43]: no nitrogen (N$_1$,0 g grove$^{-1}$), low nitrogen (N$_2$,540 g grove$^{-1}$), and high nitrogen (N$_3$,1080 g grove$^{-1}$). Urea (46% N) was applied at the end of March 2019. Urea was applied in two equal doses, with the second dose administered later in June. Throughout the experiment, the bamboo plants were consistently irrigated three to four times per week. To prevent the exchange of nutrients between treatments, ditches were dug to a depth of 50 cm along each treatment boundary

and isolated using plastic sheets (0.2 mm). Finally, the ditches are buried and compacted with soil.

Soil samples were obtained at the end of August in both 2017 and 2018, using the "LY/T 1275–1999 Forest soil analysis method" sampling method [44]. Samples were taken from the 0–20 cm and 20–40 cm soil layers, with every five sampling sites combined to form a 1 kg soil sample. Six soil samples were collected each year. Similarly, soil sampling was conducted at the end of May and August 2019 using samples from the 0 to 20 and 20–40 cm soil layers, respectively. Every five sampling sites were mixed to form a 1 kg soil sample, resulting in a total of four soil samples per treatment. In March 2019, there were 167 bamboo groves in the greenhouse, each with three to seven stems. A total of 83 bamboo groves were selected for nitrogen treatment, and the three nitrogen treatments were 29, 26, and 28, respectively. For each nitrogen treatment, 18 healthy bamboo shoots were sampled to determine dry matter and nutrient content. In addition, six bamboo groves with an age structure of "1 mother bamboo, 2 biennial young culms and 2 annual young culms" were selected for each nitrogen treatment to investigate parameters such as bamboo shoot diameter at breast height (DBH), individual weight, quality, and yield. Starting from the start of the experiment, bamboo shoots were observed weekly, and new shoot quantities were recorded during the growth period from May to August. After excavating the bamboo shoots at heights of 10–15 cm, their weight was recorded, and the diameter at breast height (DBH) was measured. The average individual weight was calculated at the end of August, and the total weight of each grove was determined. To obtain the yield parameters, the quantity of new shoots observed during the entire growing period was multiplied by the average individual weight of the shoots.

Following collection, the bamboo shoot and soil samples were brought to the laboratory for further analysis. The bamboo sheaths were promptly removed from the collected shoots, dried at 70 °C, crushed through a 0.5 mm sieve, and carefully preserved. The soil samples were air-dried, crushed through 2 mm and 0.149 mm sieves, and preserved for future analyses.

### 2.3. Methods for Determining Soil Salt Ions and Nutrients (Soil and Bamboo Shoot)

The pH meter method (soil-to-water ratio of 2.5:1) was used to measure the soil pH, and the EC value was equal to that of the conductivity meter method (soil-to-water ratio of 5:1, KCl 2.55). The exchangeable sodium (ESP) value is equal to the sodium cation exchange capacity. Ammonium acetate/ammonium hydroxide exchange flame spectrophotometry was used to determine exchangeable sodium in the soil. Cation exchange capacity was determined using the ammonium chloride/ammonium acetate exchange method. Soil organic carbon was determined via potassium dichromate oxidation and external heating. Total nitrogen (TN) was determined using the Kjeldahl method, total phosphorus (TP) was determined using the molybdenum antimony resistance colorimetric method, and total potassium (TK) was determined using alkali melt flame spectrophotometry. Alkali-hydrolyzed nitrogen (AN) was determined using the alkali-diffusion method, ammonium nitrogen ($NH_4^+$-N) using magnesium oxide leaching and diffusion, and nitrate nitrogen ($NO_3^-$-N) using the colorimetric method of phenol disulfonic acid. Available phosphorus (AP) was determined using the 0.5 mol/L sodium bicarbonate leaching method, and available potassium (AK) was determined using ammonium acetate extraction and flame spectrophotometry. Water-soluble $K^+$ and $Na^+$ concentrations were determined via flame spectrophotometry; $Ca^{2+}$ and $Mg^{2+}$ were determined via atomic absorption spectrophotometry; $HCO_3^-$ was determined via two-indicator neutralization titration; $Cl^-$ was determined via silver nitrate titration; $SO_4^{2-}$ was determined via indirect EDTA titration.

Furthermore, the nitrogen, phosphorus, and potassium contents of the bamboo shoots were determined. The nitrogen content was digested with concentrated sulfuric acid and determined using the micro-Kjeldahl method. The phosphorus content was digested with $H_2SO_4$-$H_2O_2$ and determined using the vanadium molybdenum yellow colorimetric

method. The potassium content was digested with $H_2SO_4$-$H_2O_2$ and determined using a flame photometer.

### 2.4. Statistical Analysis

Data were analyzed using ANOVA, and comparisons between treatment means were performed using the F-test at 5% probability. Generalized linear model (GML) analysis was used to evaluate the effects of stage factor ($F_S$), nitrogen application ($F_N$), and soil depth factor ($F_D$). Statistical analysis was performed using SPSS 16.0, and figures were plotted using Origin 2021.

## 3. Results

### 3.1. Growth of D. oldhami in Different Planting Years (2017–2019)

In 2017, 309 mother bamboos were transplanted, of which 114 survived, with a survival rate of 36.89%. New shoots emerged in August of that year, with a total of 0.21 stems/m² shoot quantity and 21.41 g/m² total biomass. By the end of 2017, the standing culm density in the greenhouse was 0.29 stems/m². In 2018, the total new shoot quantity was 0.92 stems/m², and the total biomass was 174.35 g/m². After the bamboo forest structure regulation, the standing culm density was 0.61 stems/m² by the end of 2018. A large bamboo shoot harvest began in 2019, the total new shoot quantity was 0.99 stems/m², and the total biomass was 274.96 g/m². After bamboo forest structure regulation, the standing culm density was 0.69 stems/m² at the end of 2019 (Table 2).

**Table 2.** The transplanting of mother bamboo, the number of bamboo shoots, and the standing culm density each year (2017–2019).

| Year | Number of Transplanting | Number of Mother Bamboo Alive | Total Number of New Shoots Stems/m² | Total Biomass of New Shoots g/m² | Standing Culm Density Stems/m² |
|---|---|---|---|---|---|
| 2017 | 309 | 114 | 0.21 | 21.41 | 0.29 |
| 2018 | - | - | 0.92 | 174.35 | 0.61 |
| 2019 | - | - | 0.99 | 274.96 | 0.69 |

The mean DBH of the transplanted mother bamboo was 31.86 mm, and the mean DBH of the newly emerged shoots first decreased and then increased annually from 2017 to 2019, with decrease/increase rates of 96.86%, 101.59%, and 35.48%, respectively. The individual weights of the new shoots also increased annually. The average weight of the new shoots in 2017 was 104.01 g, which increased by 82.54% and 46.04% in 2018 and 2019, respectively (Table 3).

**Table 3.** Diameter at breast height (DBH) and individual weight of mother bamboo and new shoots at each year (2017–2019).

| | DBH mm | Sample Number | Minimum Value | Maximum Value | Individual Weight g | Sample Number | Minimum Value | Maximum Value |
|---|---|---|---|---|---|---|---|---|
| Mother bamboo | 31.86 ± 0.58 | 114 | 17.39 | 49.11 | - | - | - | - |
| 2017 | 17.56 ± 0.38 | 247 | 5.43 | 34 | 104.01 ± 7.30 | 10 | 79.35 | 150.12 |
| 2018 | 35.40 ± 0.77 | 125 | 11.45 | 55.23 | 189.86 ± 10.31 | 20 | 116.74 | 285.7 |
| 2019 | 47.96 ± 1.19 | 20 | 34.64 | 55.34 | 277.27 ± 9.83 | 15 | 226.21 | 347.79 |

### 3.2. Soil Properties in Different Planting Years (2017–2019)

The ESP, pH, EC, water-soluble salt ions ($K^+$, $Na^+$, $Ca^{2+}$, $Mg^{2+}$, $Cl^-$, $SO_4^{2-}$, and $HCO_3^-$), and the total number of cations content significantly changed from 2017 to 2019 ($p < 0.05$, Table 4). The $Ca^{2+}$ and $HCO_3^-$ are the main cations and anions, accounting for more than 50% of the cations and anions. Cations $K^+$ and $Na^+$ and anions $Cl^-$ and $SO_4^{2-}$ increased over the planting years, while $HCO_3^-$, $Ca^{2+}$, and $Mg^{2+}$ decreased. The largest

increase was observed in K$^+$ and Cl$^-$. The proportions of Ca$^{2+}$ and HCO$_3^-$ decreased annually but remained the main cations and anions (Figure 1). Overall, the total number of cations decreased annually, whereas the number of anions increased, resulting in decreased ESP, pH, and EC. However, the EC value was not significantly different in the latter two years (2018 and 2019). The TN, C/N, AN, AP, and AK contents were significantly different ($p < 0.05$, Table 5). With increasing planting years, the TN content decreased annually, whereas AN, AP, and AK contents increased. The C/N ratio also showed an increasing trend, but the difference was not significant in the latter two years (2018 and 2019).

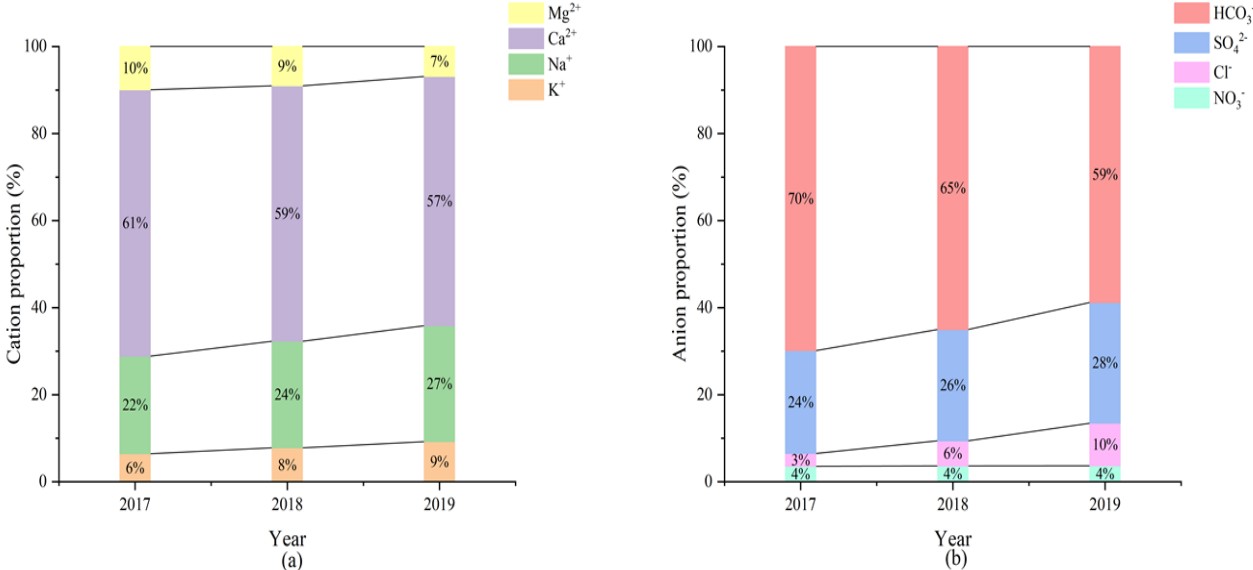

**Figure 1.** The proportion of soluble cations (**a**) K$^+$, Na$^+$, Ca$^{2+}$, and Mg$^{2+}$, and soluble anions (**b**) NO$_3^-$, Cl$^-$, SO$_4^{2-}$, and HCO$_3^-$ in total cations and total anions in soil from 2017 to 2019, respectively.

**Table 4.** Differences in soil soluble salt ions, electrical conductivity (EC), pH, and exchangeable sodium (ESP) among different planting years.

| Year | K$^+$ mg/kg | Na$^+$ mg/kg | Ca$^{2+}$ mg/kg | Mg$^{2+}$ mg/kg | Total Cation mg/kg |
|---|---|---|---|---|---|
| 2017 | 26.099 ± 0.632c | 90.540 ± 0.655c | 247.010 ± 3.196a | 40.128 ± 0.574a | 403.778 ± 4.277a |
| 2018 | 30.012 ± 0.699b | 93.911 ± 1.182b | 225.221 ± 2.909b | 34.679 ± 0.939b | 383.823 ± 5.548b |
| 2019 | 34.207 ± 0.717a | 98.790 ± 0.294a | 212.365 ± 1.904c | 25.485 ± 0.522c | 370.848 ± 2.964b |
| **Year** | **NO$_3^-$ mg/kg** | **Cl$^-$ mg/kg** | **SO$_4^{2-}$ mg/kg** | **HCO$_3^-$ mg/kg** | **Total Anion mg/kg** |
| 2017 | 18.623 ± 0.328a | 15.208 ± 1.022c | 123.100 ± 3.315c | 363.763 ± 5.664a | 520.695 ± 8.336a |
| 2018 | 19.355 ± 0.279a | 30.513 ± 0.475b | 135.300 ± 4.439b | 344.130 ± 4.806b | 529.300 ± 9.574a |
| 2019 | 19.868 ± 0.566a | 52.750 ± 0.457a | 149.695 ± 1.185a | 317.037 ± 1.238c | 539.345 ± 2.834a |
| **Year** | **ESP** | **pH** | **EC** | | |
| 2017 | 0.297 ± 0.004a | 8.263 ± 0.009a | 0.270 ± 0.005a | | |
| 2018 | 0.248 ± 0.006b | 8.228 ± 0.009ab | 0.240 ± 0.004b | | |
| 2019 | 0.175 ± 0.005c | 8.190 ± 0.027b | 0.233 ± 0.003b | | |

A significant difference at the planting years is indicated by different lowercase letters (*F* test, $p < 0.05$). Values are means ± S.E.

**Table 5.** Differences in soil nutrients among different planting years.

| Year | Organic Carbon g/kg | Total Nitrogen g/kg | C/N | Total Phosphorus g/kg | Total Potassium g/kg |
|------|---------------------|---------------------|-----|------------------------|----------------------|
| 2017 | 9.645 ± 0.191a | 1.132 ± 0.005a | 8.527 ± 0.201b | 1.104 ± 0.021a | 17.140 ± 0.053a |
| 2018 | 10.188 ± 0.378a | 1.078 ± 0.006b | 9.452 ± 0.359a | 1.136 ± 0.016a | 17.223 ± 0.050a |
| 2019 | 10.280 ± 0.275a | 1.020 ± 0.007c | 10.092 ± 0.227a | 1.095 ± 0.014a | 17.035 ± 0.010a |

| Year | Available Phosphorus mg/kg | Available Potassium mg/kg | Alkali-Hydrolyzed Nitrogen mg/kg | Ammonium Nitrogen mg/kg | Nitrate Nitrogen mg/kg |
|------|----------------------------|---------------------------|----------------------------------|--------------------------|------------------------|
| 2017 | 28.133 ± 1.049c | 146.463 ± 1.134b | 59.038 ± 1.112c | 7.447 ± 0.296a | 18.623 ± 0.328a |
| 2018 | 32.300 ± 0.667b | 152.037 ± 3.227b | 70.013 ± 2.210b | 7.198 ± 0.353a | 19.355 ± 0.279a |
| 2019 | 46.363 ± 1.343a | 163.003 ± 3.819a | 91.168 ± 1.766a | 6.388 ± 0.362a | 19.868 ± 0.566a |

Statistics as in Table 4.

### 3.3. Response of Bamboo Shoots Yield and Nutrient Content to Nitrogen Application

Nitrogen application increased DBH and individual shoot weight, but the effects were not significant ($p > 0.05$, Figure 2). The quantity and total yield significantly increased with increasing nitrogen application levels ($p < 0.01$, Figure 2). Compared to the $N_1$ level, the total yield of the $N_2$ and $N_3$ treatments increased by 33.50% and 45.22%, respectively, and the quantity increased by 23.33% and 33.33%, respectively (Figure 2).

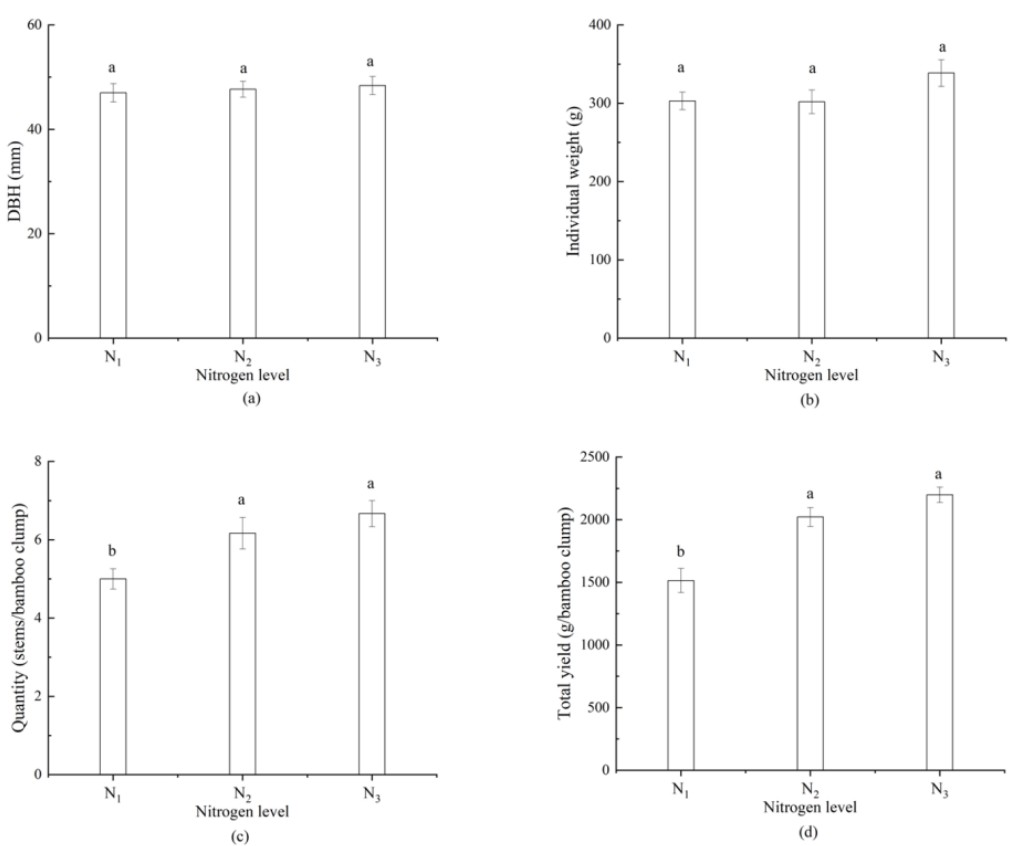

**Figure 2.** Differences in DBH (**a**), individual weight (**b**), quantity (**c**), and total yield (**d**) at different nitrogen levels during shooting stage. Different lowercase letters indicate significant differences in different nitrogen levels. Vertical bars show ± S.E. $N_1$, $N_2$, and $N_3$ refer to nitrogen application levels, which were no nitrogen, medium nitrogen, and high nitrogen, respectively.

During the shooting stage of *D. oldhami*, nutrient accumulation (specifically nitrogen and phosphorus) and nutrient concentration (specifically nitrogen, phosphorus, and potassium) of bamboo shoots were significantly affected by nitrogen levels ($p < 0.01$, Table 6). The

nitrogen and phosphorus accumulation increased with increasing nitrogen levels, but there was no significant difference between the $N_2$ and $N_3$ levels. Compared with the $N_1$ level, the nitrogen content of the $N_2$ and $N_3$ levels increased by 23.76% and 25.33%, respectively, and the phosphorus content increased by 27.36% and 30.41%, respectively. Potassium accumulation in bamboo shoots showed no significant differences among the three nitrogen levels, and $N_2$ and $N_3$ levels increased by 1.82% and decreased by 3.03%, respectively (Table 6). Compared with the $N_1$ level, the nitrogen concentrations in $N_2$ and $N_3$ increased by 14.50% and 12.82%, respectively, and the phosphorus concentration increased by 30.40% and 45.07%, respectively. The potassium content of shoots also decreased with increasing nitrogen levels, but there was no significant difference between the $N_2$ and $N_3$ levels (6.01% and 7.43%, respectively) (Table 6).

**Table 6.** Differences in nutrient content of bamboo shoots among different nitrogen levels.

| Nitrogen Level | Nitrogen Concentration g/kg | Phosphorus Concentration g/kg | Potassium Concentration g/kg |
|---|---|---|---|
| $N_1$ | 27.405 ± 0.323b | 2.658 ± 0.062c | 42.761 ± 0.247a |
| $N_2$ | 31.378 ± 0.388a | 3.466 ± 0.063b | 40.193 ± 0.236b |
| $N_3$ | 30.917 ± 0.362a | 3.856 ± 0.076a | 39.584 ± 0.256b |
| **Nitrogen Level** | **Nitrogen Accumulation g** | **Phosphorus Accumulation g** | **Potassium Accumulation g** |
| $N_1$ | 2.740 ± 0.045b | 0.296 ± 0.010b | 4.287 ± 0.097a |
| $N_2$ | 3.391 ± 0.069a | 0.377 ± 0.013a | 4.365 ± 0.130a |
| $N_3$ | 3.434 ± 0.049a | 0.386 ± 0.009a | 4.417 ± 0.113a |

A significant difference in the nitrogen levels is indicated by different lowercase letters (*F* test, $p < 0.05$). Values are means ± S.E.

### 3.4. Response of Soil Properties to Nitrogen Application

Soil ESP, pH, and EC were significantly affected by the stage, nitrogen level, and soil depth during the growing stage of *D. oldhami* ($p < 0.01$, Table 7). The ESP, pH, EC, $K^+$, $Na^+$, $Ca^{2+}$, $Mg^{2+}$, $NO_3^-$, $Cl^-$, $SO_4^{2-}$, $HCO_3^-$, and total cations all showed the highest responses to the stage factor, and then to the nitrogen level and soil depth factor ($F_S > F_N > F_D$, Table 7). Compared with the values at the shoot elongation stage, the soil ESP, pH, and EC decreased at the end of the shooting stage. With increasing nitrogen application, the ESP and pH decreased significantly, whereas EC increased (Figure 3). Water-soluble salt ions were significantly affected by several factors. At the shoot elongation stage, the EC had the highest values of 0.39 and 0.42 at $N_2$ and $N_3$ levels, and at the end of the shooting stage, the EC had the highest values of 0.26 and 0.30 at $N_2$ and $N_3$ levels. The stage factors affected $K^+$, $Na^+$, $Ca^{2+}$, $Mg^{2+}$, $NO_3^-$, $Cl^-$, $SO_4^{2-}$, $HCO_3^-$, total cations, and anions ($p < 0.01$, Table 7). The nitrogen levels affected $K^+$, $Na^+$, $NO_3^-$, $SO_4^{2-}$, $HCO_3^-$, total cations, and anions ($p < 0.05$, Table 4). The soil depth affected $K^+$, $SO_4^{2-}$, $HCO_3^-$, and total anions ($p < 0.05$; Table 7). All cations, $NO_3^-$, and $SO_4^{2-}$ contents decreased at the end of the shooting stage compared with those at the shoot elongation stage, whereas the $Cl^-$ and $HCO_3^-$ contents increased. In general, the total anions and cations decreased, anion–cation distribution shifted at the shoot elongation stage, and total cations were greater than the anions, whereas the total anions were greater than the cations at the end of the shooting stage. With increasing nitrogen application, the $Na^+$ and anion content increased, whereas the content of $K^+$ decreased. In general, the total number of anions and cations increased after the nitrogen application. The $K^+$, $SO_4^{2-}$, and $HCO_3^-$ concentrations were lower in the deep layer than in the shallow layer (Figure 4). Although the proportion of cations in each treatment remained unchanged at the shoot elongation stage ($Ca^{2+} > Na^+ > Mg^{2+} > K^+$), the proportions of $K^+$ and $Mg^{2+}$ differed at the end of the shooting stage. Similarly for anions, the proportions of $Cl^-$ and $NO_3^-$ changed between the shoot elongation stage ($HCO_3^- > SO_4^{2-} > NO_3^- > Cl^-$) and the end of elongation stage ($HCO_3^- > SO_4^{2-} > Cl^- > NO_3^-$) (Figure 5).

**Table 7.** Proportion of explained variance by stage, nitrogen level, and soil depth (generalized linear models) of soil acid–base property, salt ions, and nutrients.

| Effects | K$^+$ | Na$^+$ | Ca$^{2+}$ | Mg$^{2+}$ | Total Cation | NO$_3^-$ | Cl$^-$ | SO$_4^{2-}$ |
|---|---|---|---|---|---|---|---|---|
| F$_S$ | 286.340 ** | 73.650 ** | 63.286 ** | 891.797 ** | 233.751 ** | 142.070 ** | 814.944 ** | 2077.260 ** |
| F$_N$ | 175.210 ** | 22.551 ** | 4.539 ns | 2.656 ns | 51.926 ** | 2.593 ns | 5.782 ns | 68.407 ** |
| F$_D$ | 5.344 * | 0 ns | 0.942 ns | 0.007 ns | 1.062 ns | 0.696 ns | 2.152 ns | 29.784 ** |

| Effects | HCO$_3^-$ | Total Anion | ESP | pH | EC | Organic Carbon | Total Nitrogen | Total Phosphorus |
|---|---|---|---|---|---|---|---|---|
| F$_S$ | 304.920 ** | 39.905 ** | 290.909 ** | 248.029 ** | 2360.273 ** | 12.505 ** | 75.042 ** | 3.552 ns |
| F$_N$ | 47.718 ** | 279.072 ** | 85.364 ** | 85.590 ** | 366.903 ** | 91.366 ** | 143.025 ** | 39.445 ** |
| F$_D$ | 4.938 * | 70.430 ** | 8.909 ** | 16.116 ** | 21.569 ** | 9.860 ** | 2.255 ns | 0.716 ns |

| Effects | Available Potassium | Alkali-Hydrolyzed Nitrogen | Ammonium Nitrogen | Nitrate Nitrogen | | | | |
|---|---|---|---|---|---|---|---|---|
| F$_S$ | 20.849 ** | 155.435 ** | 81.084 ** | 142.070 ** | | | | |
| F$_N$ | 271.212 ** | 17.240 ** | 2.818 ** | 2.593 ns | | | | |
| F$_D$ | 8.695 ** | 1.008 ns | 0.848 ** | 0.696 ns | | | | |

Significant levels: ** $p < 0.01$; * $p < 0.05$; ns: non-significant ($p > 0.05$). EC: electrical conductivity; ESP: exchangeable sodium.

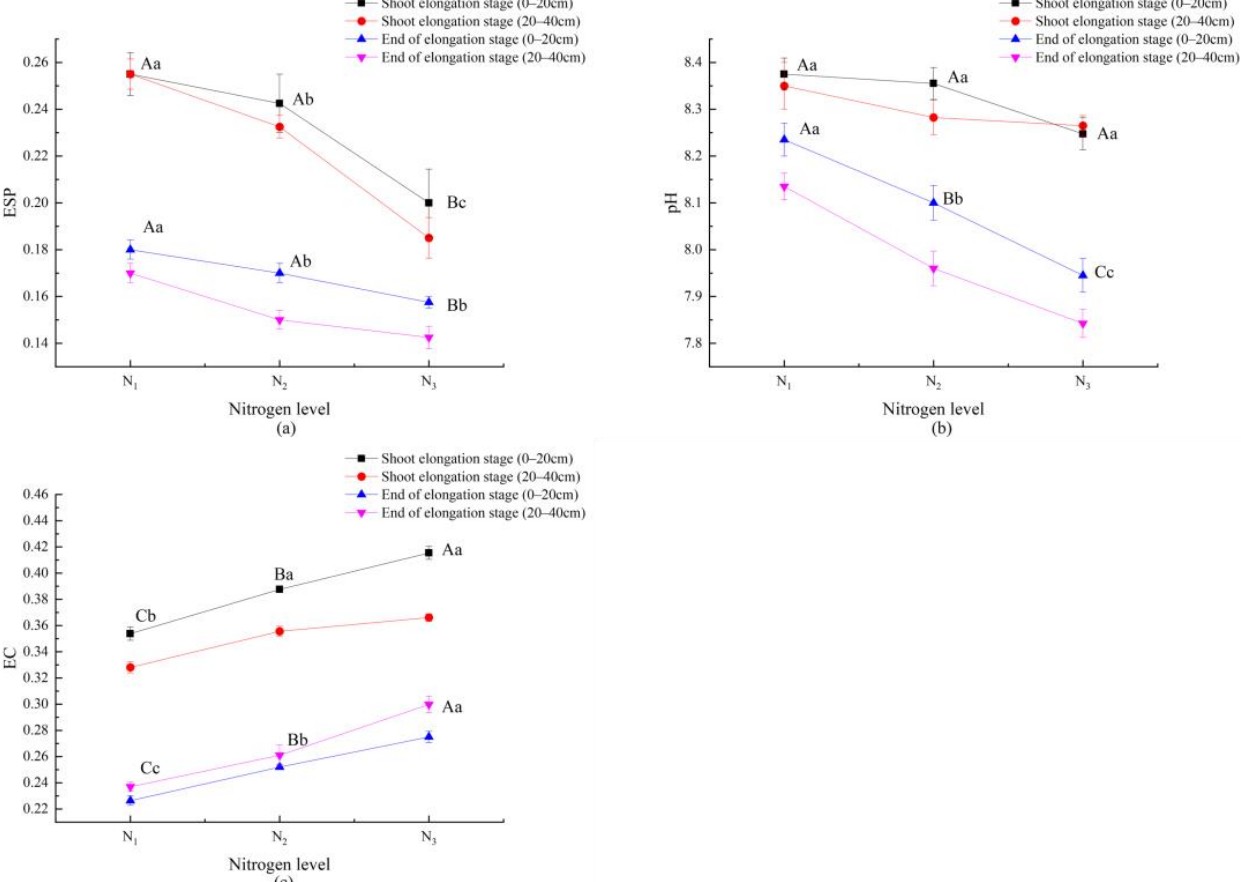

**Figure 3.** Differences of exchangeable sodium (ESP) (**a**), pH (**b**), and electrical conductivity (EC) (**c**) in soils (0–20 cm and 20–40 cm soil layers) at different nitrogen levels during shoot elongation stage and the end of elongation stage. A significant difference at the nitrogen levels is indicated by different capital letters for 0–20 cm (shoot elongation stage and the end of elongation stage) and by different lowercase letters for 20–40 cm (shoot elongation stage and the end of elongation stage) (*F* test, $p < 0.05$). Vertical bars show ± S.E. N$_1$, N$_2$, and N$_3$ refer to nitrogen application levels, which were no nitrogen, medium nitrogen, and high nitrogen, respectively.

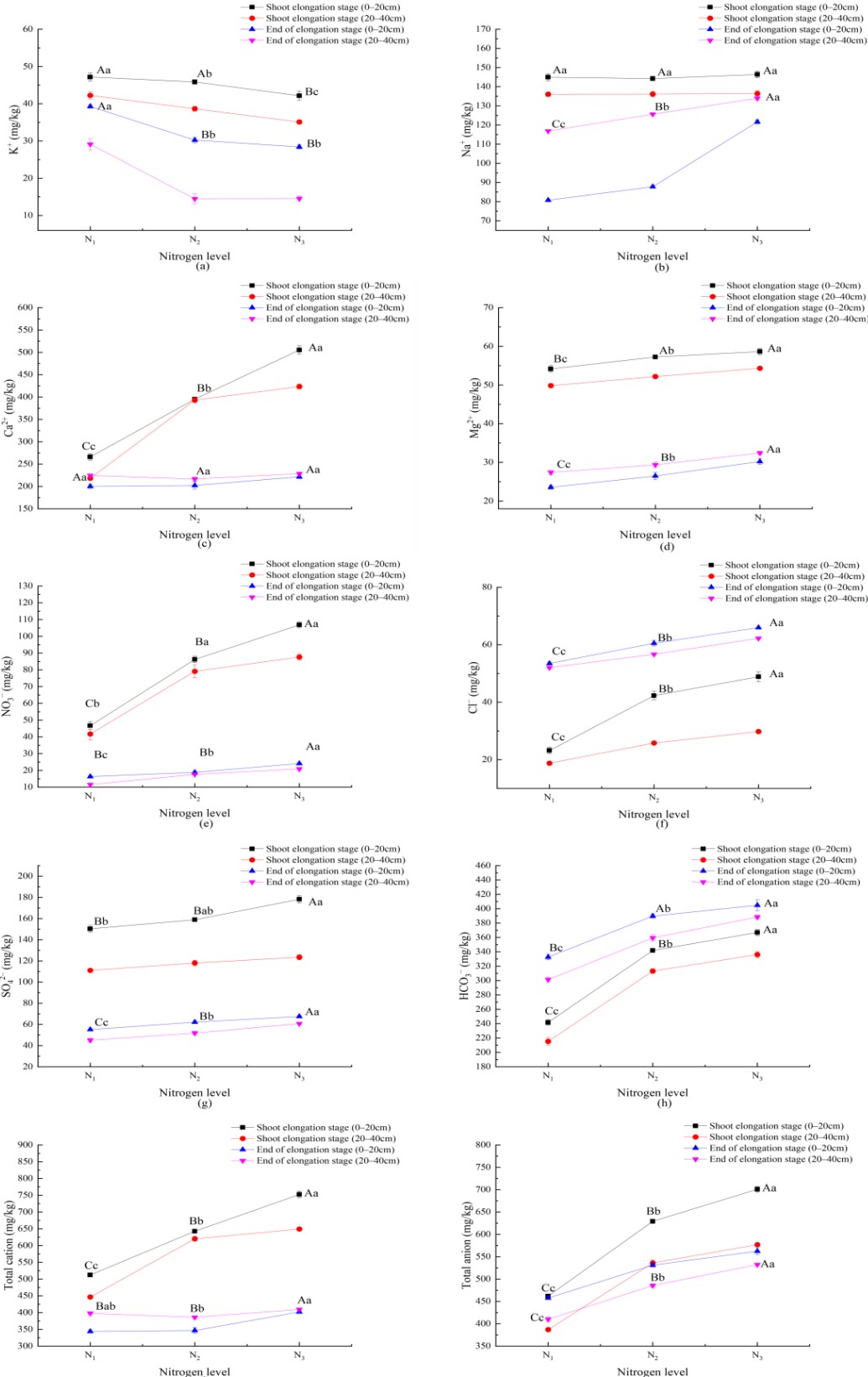

**Figure 4.** Differences in water-soluble cation K$^+$ (**a**), Na$^+$ (**b**), Ca$^{2+}$ (**c**), Mg$^{2+}$ (**d**), water-soluble anion NO$_3^-$ (**e**), Cl$^-$ (**f**), SO$_4^{2-}$ (**g**), HCO$_3^-$ (**h**), total cations (**i**), and total anions (**j**) in soils (0–20 cm and 20–40 cm soil layers) at different nitrogen levels during shoot elongation stage and the end of elongation stage. Statistics are similar to those in Figure 3.

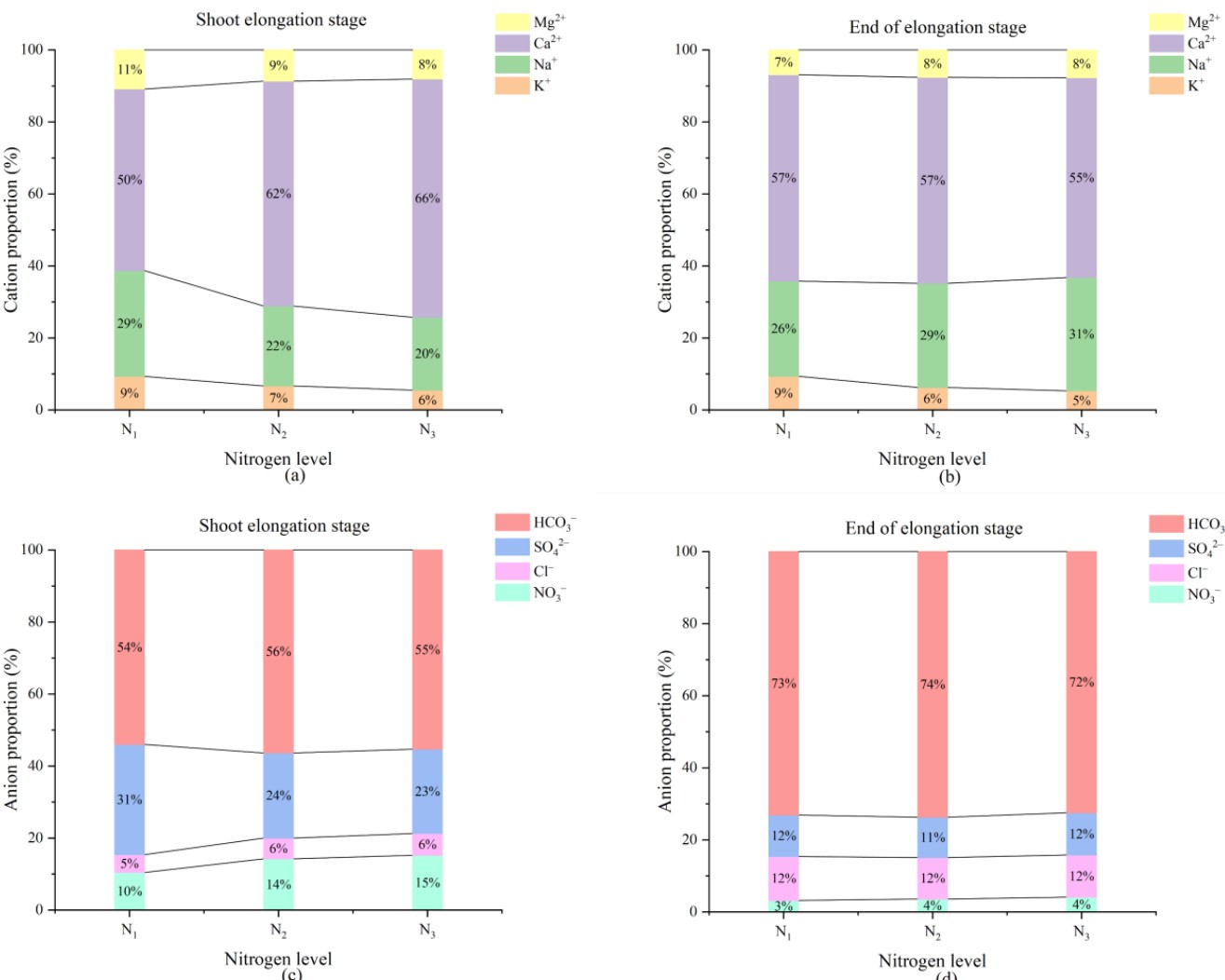

**Figure 5.** The proportion of soluble cations (K$^+$, Na$^+$, Ca$^{2+}$, and Mg$^{2+}$) during shoot elongation stage (**a**) and the end of elongation stage (**b**), and soluble anions (NO$_3$$^-$, Cl$^-$, SO$_4$$^{2-}$, and HCO$_3$$^-$) during shoot elongation stage and the end of elongation stage in total cations and total anions in soil at different nitrogen levels during shoot elongation stage (**c**) and the end of elongation stage (**d**).

During the growth stage of *D. oldhami*, the soil nutrient content was significantly influenced by several factors. The stage factor affected the soil nutrient content (except for TP) ($p < 0.01$; Table 7). Nitrogen levels affected soil nutrient content (except for ammonium nitrogen and nitrate nitrogen) ($p < 0.01$; Table 7). Soil depth affected soil organic carbon, AP, and AK ($p < 0.01$, Table 7). Soil AN, NH$_4$$^+$-N, and NO$_3$$^-$-N responded the most to the stage factor, followed by nitrogen level and soil depth factor (F$_S$ > F$_N$ > F$_D$, Table 7). Soil AN, NH$_4$$^+$-N, and NO$_3$$^-$-N responded the most to the nitrogen level, and the soil depth and stage factors were as follows: F$_N$ > F$_D$, F$_S$ (Table 7). The soil nutrient content at the end of the shooting stage was significantly lower than that at the shoot elongation stage. Soil organic carbon, TN, TP, TK, AP, and AK content also decreased with nitrogen application. Nitrogen application significantly increased the soil AN content at the shoot elongation stage but decreased the soil AN content at the end of the shooting stage. Soil organic carbon, AP, and AK accumulated easily in the shallow layers (Figure 6).

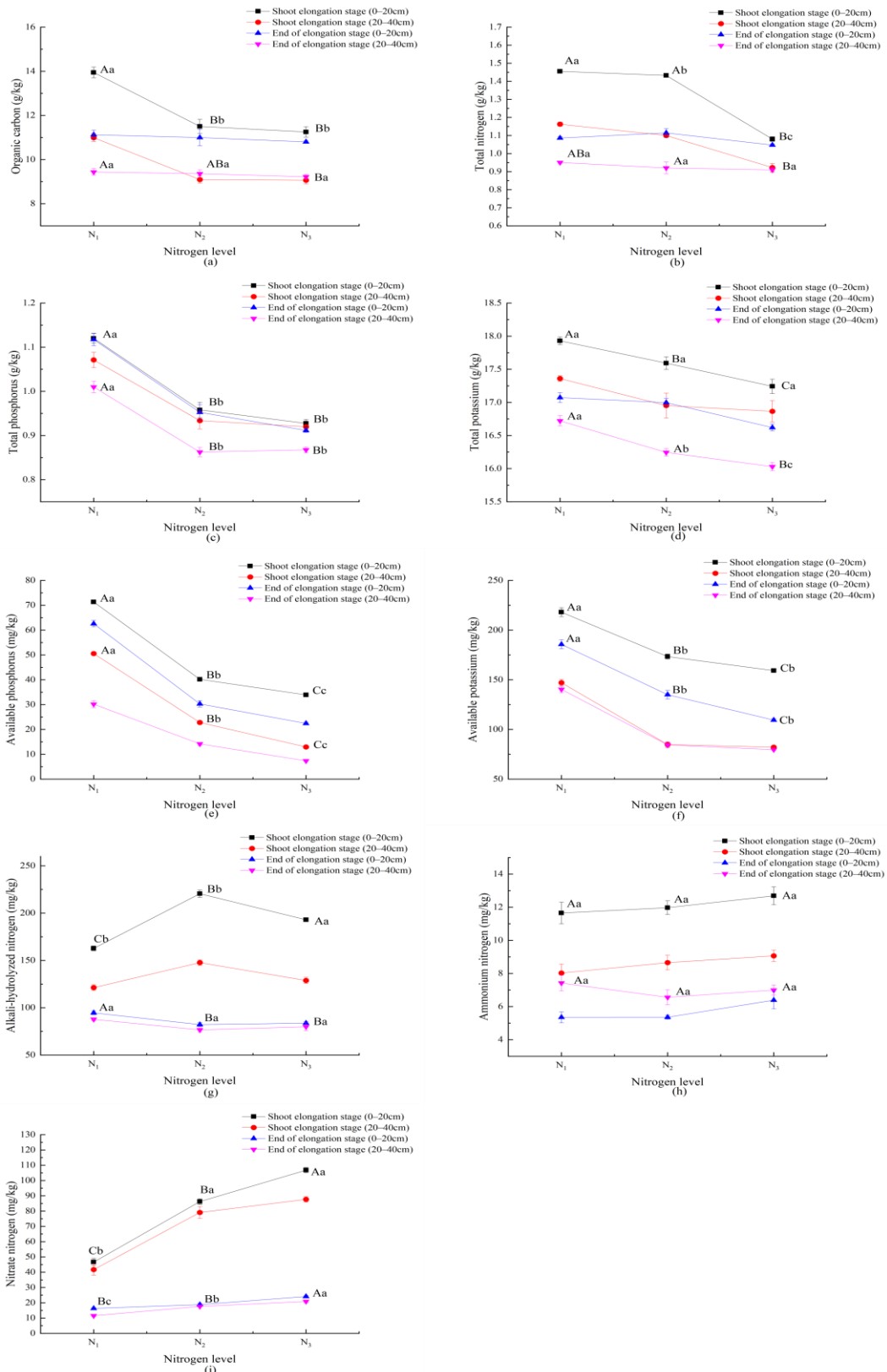

**Figure 6.** Differences of organic carbon (**a**), total nitrogen (**b**), total phosphorus (**c**), total potassium (**d**), available phosphorus (**e**), available potassium (**f**), alkali-hydrolyzed nitrogen (**g**), ammonium nitrogen (**h**), and nitrate nitrogen (**i**) in soils (0–20 cm and 20–40 cm soil layers) at different nitrogen levels during shoot elongation stage and the end of elongation stage. Statistics are similar to those in Figure 3.

## 4. Discussion

### 4.1. Effects of Planting Years on Soil Saline–Alkali Properties and Nutrient Changes

After the introduction of *D. oldhami* to greenhouses in northern China, the DBH and quantity of bamboo shoots decreased in 2017. However, significant growth was observed in the subsequent year (2018), with an explosive increase in the quantity, total biomass, and standing culm density of bamboo shoots. In the third year (2019), bamboo shoots were successfully harvested. Compared to the study by Jin et al. [39], the survival rate (36.89%) of *D. oldhami* was lower after it was introduced into a greenhouse in northern China, and the quantity of bamboo shoots produced each year was also lower. This may be attributed to the need for *D. oldhami* to adapt to local soil properties and the challenges posed from increased latitudes to combat climate change. Additionally, fluctuations in $CO_2$ concentrations and delayed artificial irrigation in greenhouse environments may also affect bamboo shoot growth [45,46]. Overall, this study aligns with the process of forest formation observed in the first 3 years after the introduction of *D. oldhami* to coastal mudflats by Jin et al. [39], as it achieved successful introduction and survival. Analysis of the salt content in greenhouse soil using the cation and ion addition method [44] revealed a range of 0.09–0.15% (Table 4), which was significantly lower than the 0.41% threshold observed in coastal mudflat soil in Jin et al.'s study [39]. Therefore, soil salinity is unlikely to impede the survival of *D. oldhami* under greenhouse conditions. However, whether it hinders its nutrient absorption efficiency remains to be determined. Although the salinity and ion composition of greenhouse soils vary, many similarities exist [18,47]. Some studies have reported that $NO_3^-$ and $SO_4^{2-}$ are among the main characteristics of secondary salinization of soil in greenhouses [15,48–50], whereas $Cl^-$ and $SO_4^{2-}$ are the main anions, and $Ca^{2+}$ and $K^+$ are the main cations [51,52]. Differences in the type and quantity of soil salt ions in the greenhouse led to differences in the soil surface charge, soil pH, and ESP. Therefore, the secondary salinization of soil is closely related to soil acidification and exchangeable bases [24,53,54]. In this study, $Ca^{2+}$ and $HCO_3^-$ were the main cations and anions in greenhouse soil, accounting for more than half of the total cations and anions, respectively. $HCO_3^-$ can be hydrolyzed to produce $OH^-$, which is an important cause of soil alkalization. After the introduction of *D. oldhami*, the proportion of $Ca^{2+}$ and $HCO_3^-$ was reduced but remained dominant (Figure 1). The main cause of soil acidification was the increasing proportions of $NO_3^-$ and $SO_4^{2-}$ in the total salt content [55]. Some studies have also suggested that the significant increase in $NO_3^-$, $SO_4^{2-}$, and $Cl^-$ in the soil, as well as the decrease in the relative proportion of the soil cations $Ca^{2+}$, $Mg^{2+}$, and $Na^+$, are important for the decrease in soil pH [53,56]. Our results agree with these studies, showing that a decrease in soil cations and an increase in anions were the primary causes of salinized soil acidification (Table 4). From the perspective of plant invasion, it has been found that the invasion of *Phyllostachys edulis* increases soil pH in subtropical forests in China [31] and decreases soil pH in Jiangxi Province in southeastern China [57]. However, the reason for this remains unclear. In this study, the ESP and pH of greenhouse soil from *D. oldhami* transplanted from south to north decreased annually, which was related to the increase in total anions and decrease in total cations each year (Table 4). This may also be related to the effects of the bamboo root systems, which secrete various organic acids in heterogeneous environments based on soil acidity and alkalinity [58,59]. Bamboo invasion positively affected the physical properties of soil [60]. The soil nitrogen and carbon cycles are important processes mediated by microorganisms. Any variation is often related to changes in soil acidity and alkalinity, and there is often a significant correlation with available nutrient content [61–63]. Therefore, the greenhouse soil available nutrients (AN, AP, and AK) were activated in the north-transplanted bamboo (Table 5). However, the *D. oldhami* forest expanded rapidly in the following year (2018), and the standing culm density doubled. The third year (2019) ushered in a bamboo shoot harvesting year. The quantity, DBH, and individual weight increased annually. Therefore, bamboo shoot emergence and growth absorbed large amounts of nutrients from the soil (Tables 2 and 3). Litter and other inputs were lower than the output, especially nitrogen, and the TN content significantly

decreased each year [64,65], which is consistent with the results of this study. There were no significant changes in organic carbon, TP, or TK (Table 5), which were in equilibrium with the input of litter or soil mineralization and the output of bamboo absorption. The C/N ratio increased annually but was lower than the average global soil C/N ratio (13.11) [66]. This reflects the rapid decomposition rate of organic matter and the relative increase in soil fertility. In addition, it has been reported that the ratio of $NH_4^+$-N and $NO_3^-$-N in the soil often increases after the bamboo invasion of evergreen broadleaf forest [67]. However, the calcareous soil in this study exhibited an alkaline reaction and vigorous nitrification from $NH_4^+$-N to $NO_3^-$-N. The $NO_3^-$-N was the dominant nitrogen source, and the introduction of *D. oldhami* did not change this trend (Table 5).

### 4.2. Effects of Nitrogen Application on Soil Saline–Alkali Properties, Soil Nutrients, and Bamboo Shoots Nutrients during the Growing Stage

The anion and cation contents of the soil changed with seasonal changes. In general, the total cation content at the shoot elongation stage was higher than the anion content, and the total anion content at the end of the shooting stage was higher than the cation content (Figure 4). This caused the acidification of greenhouse soils in summer and a decrease in pH and ESP (Figure 3). The effects of nitrogen application on the anions and cations differed. Although nitrogen application had little effect on the proportion of anions, it increased the content of $Na^+$, $Ca^{2+}$ (shoot elongation stage), and $Mg^{2+}$ (Figure 4). This is related to the exchange of cations adsorbed by $NH_4^+$ and soil colloids after nitrogen application [68]. However, $NO_3^-$ is not easily adsorbed by the soil solid phase and exists in the soil solution. Under electrochemical equilibrium, the soil solution should maintain electrical neutrality and promote the dissociation of $Na^+$, $Ca^{2+}$, $Mg^{2+}$, and other cations into the solution in equal quantities [68,69]. The increase in $Na^+$, $Ca^{2+}$, and $Mg^{2+}$ concentrations in the soil solution was also one of the reasons for the significant increase in EC after nitrogen application (Figures 3 and 4). Many studies have shown that compared with other cations, nitrogen application has a less obvious effect on the content of $K^+$ [52,70,71]. In addition, only the proportion of $Ca^{2+}$ increased significantly after nitrogen application at the shoot elongation stage, and the proportions of $Na^+$ and $K^+$ decreased. However, it recovered to a stable state at the end of the elongation (Figure 5). Possibly because of the increase in soil salt content (EC) during the shoot elongation stage and nitrogen application (Figure 3), $Ca^{2+}$ on the plasma membrane is replaced by Na+, and the enzyme-binding calcium ions lose their activity. In this study, it was observed that all soluble salt ions accumulated in the shallow layer throughout all stages, except for $K^+$ and $NO_3^-$ at the end of the elongation stage (Figure 4). As $NO_3^-$ is negatively charged, it is not easily absorbed by soil colloids and migrates to the deep layer under the drive of irrigation [72,73]. In contrast, $K^+$ and $NO_3^-$, which have opposite charges, have a synergistic effect; thus, the active absorption of $NO_3^-$ by plants can promote the absorption of $K^+$ [74]. Shallow soil nutrients are absorbed by plants quickly, so the contents of $K^+$ and $NO_3^-$ in shallow soil are lower at the end of the shooting stage.

From May to August, the shooting stage of *D. oldhami* was the most active period for bamboo to absorb nutrients. High summer temperatures and rainfall increase the soil temperature and humidity. This not only increases the number of microorganisms, metabolic activity, and root exudates [75] but is also the most important environmental factor affecting nitrogen mineralization [76,77]. Therefore, the contents of all soil nutrients decreased in August (Figure 6). Seasonal changes in the available soil nitrogen content in bamboo forests are mainly caused by seasonal changes in soil temperature and moisture. Alike the seasonal variation of inorganic nitrogen content in evergreen broad-leaved forests, soil $NH_4^+$-N was the highest in spring and decreased in summer, and $NO_3^-$-N was the lowest in summer [78]. Nitrogen application reduced the soil pH and ESP, which decreased with an increase in the nitrogen application rate (Figure 3), consistent with the results of many previous studies. It also improves the availability of soil nitrogen, changes the soil carbon cycle, and significantly reduces soil organic carbon mineralization, which decreases

with increasing nitrogen application rates [79]. Nitrogen application can promote the absorption of two kinds of inorganic nitrogen, especially $NO_3^--N$ (Figure 6). With an increase in the nitrogen level, the $NO_3^--N$ content did not show a uniform trend, which was related to the efficiency of the nitrogen input and output [80,81]. The application of nitrogen increased the $NO_3^--N$ content and promoted the absorption of nitrogen from the soil by bamboo. However, the effect of nitrogen application on the quality of the bamboo shoots during the growing period was not significant. Increases in DBH and individual weight were not obvious, but the quantity and total yield of the bamboo shoots increased significantly (Figure 2). In the two stages of soil sampling, the $NO_3^--N$ content of $N_3$ level was higher than that of $N_2$ (Figure 6), but there was no significant difference between $N_2$ and $N_3$ levels in bamboo quantity, total yield, nitrogen, and phosphorus accumulation or concentration (Figure 2, Table 6). The interaction between nitrogen and phosphorus was the most significant; this interaction has previously been shown to be beneficial [82,83]. The results indicate that the nitrogen and phosphorus uptake efficiencies of $N_2$ were higher than those of $N_3$ during the shooting stage. Based on the above similar results for bamboo shoot quantity and yield (Figure 2), as well as the response of EC to nitrogen application (Figure 3), we have concluded that nitrogen application would increase soil EC. The soil EC value between $N_2$ and $N_3$ (0.26–0.42) may be the critical value affecting the nutrient uptake of *D. oldhami*. Some studies report that there is a positive interaction between N and K absorption [84–86]. In fact, the interaction between N and K is complex [87] and is related to plant N and K starvation and soil supply [88]. The soil K supply was sufficient in this experiment (Table 5); therefore, the amount of nitrogen applied did not significantly affect K accumulation in the bamboo shoots (Table 6). The average K concentration in bamboo shoots decreased with nitrogen fertilization because the dry matter weight of the bamboo shoots increased with nitrogen application. When the K content in the soil was high, further application of nitrogen did not promote the absorption of soil K by *D. oldhami*. The main source of soil organic carbon is plant residue, which first enters the soil surface and is the main reason for SOC accumulation of soil organic carbon (Figure 6). The accumulation of available P and K in the surface layer is related to surface polymerization and eluviation of plants (Figure 6) [89,90].

## 5. Conclusions

After the introduction of *D. oldhami* at high latitudes, greenhouse soil properties changed significantly within three years, which improved the soil saline–alkali environment. The $Ca^{2+}$ and $HCO_3^-$ are the main water-soluble salt ions in greenhouse soils. The total anion content increased annually, whereas the total cation content decreased annually, resulting in decreases in ESP, pH, and EC. The salinization characteristics of the calcareous soil were alleviated, available nutrients in soil were activated, and the organic matter mineralization rate and soil fertility increased. The quality and yield of bamboo shoots increased annually, and the standing culm density increased by 1.4 times, which resulted in a significant decrease in the TN content each year.

Nitrogen application can promote the absorption of inorganic nitrogen in the form of $NO_3^--N$. However, soil salinity also increased with the increase in the nitrogen application rate, and the absorption efficiency of nitrogen and phosphorus at the $N_3$ level was lower than that at the $N_2$ level. The quantity of bamboo shoots and total yield showed similar results. Therefore, when the soil salt concentration in the greenhouse was 0.26 < EC < 0.42, the nutrient absorption of *D. oldhami* was affected. Compared with the shoot elongation stage, pH, ESP, and EC, all soil nutrient contents decreased significantly at the end of elongation.

**Author Contributions:** F.G. designed this study and revised the manuscript. Z.Y. wrote the first draft of the manuscript and performed the data analysis. Z.Y., X.Z. (Xiao Zhou) and X.Z. (Xuan Zhang) performed fieldwork. D.F. improved English language and grammar. L.L. and Z.L. provided guidance and methodological advice. All authors have read and agreed to the published version of the manuscript.

**Funding:** This research was supported by the Basic Scientific Research Funding of the International Center for Bamboo and Rattan (1632020023).

**Data Availability Statement:** Data is contained within the article. The data presented in this study are available in [insert article].

**Acknowledgments:** We thank the Fujian Academy of Forestry for providing quality bamboo material to *D. oldhami* and Chen Guobiao for their guidance and assistance with the introduction and planting work.

**Conflicts of Interest:** The authors declare no conflict of interest.

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
