# Peer review of "Soil Nutrient, Salinity, and Alkalinity Responses of Dendrocalamopsis oldhami in High-Latitude Greenhouses Depending on Planting Year and Nitrogen Application"

_forests, doi:10.3390/f14061113_

Round 1

Reviewer 1 Report

The research on the transplanting of Dendrocalamopsis ordhami was a novel concept, and your results had an originality in Forests. However, greater part of your data was soil analysis. In the present circumstances, your manuscript was unsatisfied because of poor growth data. If you want to resubmit to Forests, you should add the data of growth of D. oldhami. For example, number, height, diameter and biomass of shoot of D. oldhami should add because relationship between growth of bamboo and uptakes of nutrients was important to reflect your soil data. If you could not add the data of growth, it was better to reject. I also indicate the problem of your manuscript as following:

1. In the first paragraph of introduction, first four sentences (L. 42-47) need quotation, respectively.

2. I could not understand “callitic soil” in L. 59. You must revise this word.

3. Explanation on the results of quotation of 41 (Jin et al. 1997) was important because the results were the base of your research. However, many readers could not understand the results because this quotation was written in Chinese. You must explain the results of Jin et al. 1997 in detail.

In your introduction, you should clarify general growth characteristics of D. oldhami by transplanting. You also clarify unclear characteristics of D. oldhami to emphasize the objective of your research.

4. In L. 105, latitude and longitude were contrary.

5. On the fertilization treatment, I could not understand the reason why you add only nitrogen. Moreover, I could not understand the reason why you selected urea and its amount for the cultivation of bamboo. You must make clear the concept of fertilization. Moreover, you wrote excessive application of nitrogen fertilization in the introduction (L, 75-76). I felt that your fertilization treatment and description of excessive application was contradict. You should not mention the problem of excessive application in the introduction.

6. I could not understand the S-type sampling. You should quote the method of sampling.

7. On the analysis of bamboo shoot, there was no information of digestion. You must add this information.

8. The words on the growth period of bamboo was not general. I could not imagine the meaning of “gestating shoot stage”. You must change into “shoot elongation stage”. On the word of “end of shooting”, you should change into “end of elongation”.

9. In Table 4, there was no information in words of FS, FN and FD. You must explain in the Materials and Methods.

10. In the discussion of 4.1, you discussed the decrease of cation. You made ditches in your experiment, and there was a possibility that leaching of cation to ditches might be occurred by watering. You should explain the effects of leaching.

11. In the discussion, comparison with the results of Jin et al. 1997 was important. At first, you should show the salt content calculated by their methods. Moreover, you should compare the data of biomass.

12. On the description of absorption of nutrients (L. 349-354), you must quote the concentration of nutrients in bamboo shoot (Table 5). Moreover, you mentioned the reduction of concentration of P and K in bamboo shoot (396-397). In my opinion, calculation of content (concentration x biomass) should try to estimate the absorption of nutrients. However, you did not analyze concentration of Ca in bamboo shoot. If you have a data of concentration of Ca in bamboo shoot, this result may be important.

13. The entire description of conclusion needed revise because the contents were vague. You have to condense the important results, and concise the text. You must remove the description of (p <0.05). Moreover, the data of EC written in L. 434 was the first appearance. You must move to the results. Of course, you have to add the sentence of “conclusion”.

Reviewer 2 Report

This paper is meaningful to promote the bamboo industry in north of China. This paper aim to evaluate the effects of planting years and nitrogen fertilizer levels on salinity and soil fertility of Dendrocalamopsis oldhami forest land, which were planted at Beijing. In this paper, the soil acid-base properties, nutrients, and salt content of D. oldhami in a high-latitude greenhouse, were detected by authors. This was to provide a theoretical basis for scientific soil management in greenhouses and sustainable management of D. oldhami at Beijing. While, there were still a few questions and suggestions to this manuscript as follows.

1. According to the abstract description, it is suggested that the soil acid-base properties in the abstract match the title.

2. Since it is in the background of southern bamboo northward migration, it is suggested to add the technical difficulties of southern bamboo northward migration, and the great significance of promoting the solution of the technical difficulties through this study.

3. Pay attention to formatting details. For example, 112, L124 D. oldhami should have italics, Spaces between digits and characters, etc. It is suggested that the whole text should be checked and edited.

4. The messages of materials should be complete, such as L124 organic fertilizer and other detailed manufacturers.

5. Pay attention to grammar, e.g. L126 “set up... levels”, L129” WATERRING OCCURRED... times”" is not quite suitibale.

6. Is the quantity of test materials cultivated in large quantities in the previous cultivation? Those with consistent growth are selected as the research object in this experiment? And 6 clusters per treatment? it might be misunderstood that there are only six bushes from the beginning.

7. planting years was used as section titles in L184. What’s the meaning? Do you want to express the response of the primary parameter to the years changes?

8. NO3-–N was the dominant nitrogen source,But the experiment used urea.Is there any contradiction?

9. L334 mentioned seasonal changes, the investigation does not contain the content of seasonal changes. please reconsider.

Round 2

Reviewer 1 Report

Your manuscript was improved according to my suggestion. Especially, you added the growth data of bamboo. In the present circumstances, your manuscript was acceptable in progress. However, I would like to request further improvement on the text of discussion. On the first reason, you added some new data; however, you did not discuss on the content of new data. In my opinion, you should discuss all of data shown by Figures and Tables. On the second reason, comparison of other research article was important to emphasize the new findings of your research. You already answered on the comparison of Jin et al. 1997. However, the text of comparison did not add in the discussion. You should add the content of response 12 in the discussion. If possible, you should add on the comparison of other similar researches of D. oldhami.

I also indicate the points of the revision in your manuscript as following:

1. You must add the reason why you chose urea for the nitrogen fertilizer.

2. You must quote the literature written the sentence of “S-type” sampling.

3. In Table 6, the data was changed into “content”. However, you also need the data of concentration, which was the base on the calculation of content. You have to add the data on the concentrations of nutrients in banboo shoot.
